# Tigecycline Therapy for Infections Caused by Extended-Spectrum β-Lactamase-Producing *Enterobacteriaceae* in Critically Ill Patients

**DOI:** 10.3390/antibiotics9050231

**Published:** 2020-05-05

**Authors:** Wen-Liang Yu, Nan-Yao Lee, Jann-Tay Wang, Wen-Chien Ko, Chung-Han Ho, Yin-Ching Chuang

**Affiliations:** 1Department of Intensive Care Medicine, Chi Mei Medical Center, Tainan 710, Taiwan; Yuleon_md@yahoo.com.tw; 2Department of Medicine, School of Medicine, College of Medicine, Taipei Medical University, Taipei 100, Taiwan; 3Department of Internal Medicine and Center for Infection Control, National Cheng Kung University Hospital, Tainan 710, Taiwan; nanyao@mail.ncku.edu.tw (N.-Y.L.); winston@mail.ncku.edu.tw (W.-C.K.); 4Department of Medicine, College of Medicine, National Cheng Kung University, Tainan 710, Taiwan; 5Department of Internal Medicine, National Taiwan University Hospital, Taipei 100, Taiwan; wang.jt1968@gmail.com; 6Institute of Infectious Diseases and Vaccinology, National Health Research Institutes, Zhunan, Miaoli 350, Taiwan; 7Department of Medical Research, Chi Mei Medical Center, Tainan 710, Taiwan; ho.c.hank@gmail.com; 8Department of Hospital and Health Care Administration, Chia Nan University of Pharmacy & Science, Tainan 717, Taiwan

**Keywords:** Enterobacteriaceae, ESBL, carbapenem resistance, SOFA score, tigecycline

## Abstract

We aimed to evaluate tigecycline on the clinical effectiveness in treating complicated skin and soft tissue infections (cSSTI), complicated intra-abdominal infections (cIAI), and pneumonia, caused by extended-spectrum β-lactamase (ESBL)-producing Enterobacteriaceae, as data are limited. From three medical centers in Taiwan, we retrospectively studied the cSSTI, cIAI, and/or pneumonia caused by ESBL-producing Enterobacteriaceae. Among the 71 patients, including 39 patients infected with *Klebsiella pneumoniae*, 30 infected with *Escherichia coli* and others, the clinical success rate of tigecycline-based therapy was 80–90% for pneumonia and cSSTI caused by *E. coli* and 50–60% for cIAI caused by *K. pneumoniae* and *E. coli*. Microbiological and clinical outcome of pneumonia caused by carbapenem-resistant *K. pneumoniae* was poor. Univariate Cox analysis showed that dyspnea, SOFA score, septic shock, thrombocytopenia, prolonged prothrombin time, and lesser microbiological eradication were significant factors associated with 30-day mortality after the end of therapy. Cox regression proportional hazards model revealed dyspnea and a SOFA score > 8 to be independently associated with time to death. For ESBL producers, tigecycline showed good effects for cSSTI and pneumonia by *E. coli*, ordinary for cIAI, but ineffective for pneumonia by *K. pneumoniae*. Dyspnea and a high SOFA score predict a poor outcome.

## 1. Introduction

Tigecycline is a glycylcycline antibiotic with a broad-spectrum antimicrobial activity against multidrug-resistant organisms (MDROs), such as methicillin-resistant *Staphylococcus aureus*, extended-spectrum β-lactamase (ESBL)-producing Enterobacteriaceae, *Acinetobacter baumannii* groups and even those with carbapenem resistance [1,2]. The activity of tigecycline against MDROs remains stable over time [3].

However, the in vitro activity does not always guarantee an adequate clinical response [4]. Tigecycline is approved for the treatment of complicated skin and soft tissue infections (cSSTI), complicated intra-abdominal infections (cIAI), and community-acquired bacterial pneumonia [5]. Carbapenems have been recommended as the drug of choice for severe infections caused by ESBL producers. Therefore, clinical data remain limited regarding tigecycline used for the treatment of the infections caused by ESBL-producers [5,6].

In Taiwan, ESBL production among isolates from cIAI was common [7]. The prevalence of ESBL-producing *Escherichia coli* in the community settings has increased from 4.0% to 10.7% from 2002 to 2012 [8]. Prior hospitalization, antimicrobial use, and residence in long-term care facilities were presumed associated factors [8,9]. Moreover, carbapenem has been recognized as one of the main risk factors for the emergence of carbapenem resistance [10,11]. Other antibiotic agents, such as fluoroquinolones or β-lactam/β-lactamase inhibitors, are promising for the infections caused by ESBL producers [12,13]. However, these compounds also increasingly contributed to carbapenem resistance among ESBL-producing *Klebsiella pneumoniae* population and correlated with the rise in cefotaxime-resistant *E. coli* in Taiwan [13,14,15].

There are country-wide antimicrobial stewardship programs (ASP) available in Taiwan, implementing strategies and audit of antimicrobial use through hospital inspection and national health insurance payment system [16]. ASP systems were beneficial to reduce antibiotic consumption and antimicrobial resistance rate [17,18,19,20,21,22]. However, ESBL prevalence remained substantially high and that would lead to the increased use of carbapenems by ASP recommendations [23,24]. Therefore, challenges emerge for effective measures in ASP to restrict carbapenem use in ESBL-related infections and thus reduce carbapenem resistance [25,26].

Tigecycline may be investigated to play a potential role in the alternative strategy to treat ESBL producers [27,28]. The primary goal of this study was to evaluate the effect of tigecycline-based therapy for patients with infections caused by ESBL-producing *Enterobacteriaceae* in clinical practice in Taiwan. The second goal of the study was to analyze the clinical characteristics predictive of mortality of the patients with these serious infections.

## 2. Methods

### 2.1. Study Design

We performed a retrospective, non-comparative, and multicenter cohort study to review the charts of cases infected by ESBL-producing Enterobacteriaceae from 1 January 2015 to 31 October 2016. ESBL phenotype was reported by the local microbiology laboratory of the participated hospitals, according to guidelines by the Clinical and Laboratory Standard Institute [4]. The targeted isolates susceptible to carbapenems (including ertapenem, imipenem, and meropenem, designed ESBL-CS) or resistant to carbapenems (including ertapenem, imipenem or meropenem, designed ESBL-CR) were enrolled in the study. However, the isolates with carbapenem resistance testing negative for the ESBL phenotype were not enrolled. During the study period, tigecycline test was not routinely performed for ESBL-producing isolates, but was optional upon a physician request.

### 2.2. Study Patients

Patients of age ≥20 years, including immunocompetent and immunocompromised status with cSSTI, cIAI, and/or pneumonia, were analyzed. We enrolled pneumonia of community-acquired and nosocomial infections because ESBL-producers were common etiologies. Although not approved indications, urinary tract infection (UTI) was also enrolled along with secondary bacteremia, if any.

All patients were to have ESBL-producers, which were susceptible to tigecycline and were isolated from cultures of appropriate specimens of wound pus, ascites/intra-abdominal pus, sputum/endobronchial aspirate, and urine samples for cSSTI, cIAI, pneumonia, and UTI respectively. Reports of infectious agents were enforced by laboratory regulations according to the national ASP task force [16].

Cases of cSSTI, cIAI, pneumonia, and UTI were defined according to routine clinical practice and previously published clinical trials [5,6,27,28]. Patients with multiple infections (two or more diseases of the cSSTI, cIAI, pneumonia, and UTI) were also enrolled.

Patients received an initial dose of 100 mg of tigecycline (Pfizer Pharma GmbH, Berlin, Germany), followed by 50 mg tigecycline every 12 h, either as monotherapy or in combination with other antibiotics (piperacillin-tazobactam, fluoroquinolones, broad-spectrum cephalosporins, or carbapenems), for >72 h. The cases with concurrent fungemia or bacteremia other than *Enterobacteriaceae* were excluded. Source control and surgical intervention especially for patients with cIAIs or necrotizing cSSTIs were conducted according to routine clinical practice.

### 2.3. Study Centres

We have retrospectively evaluated the clinical and microbiological responses of patients from 3 medical centers in northern and southern areas of Taiwan. Our investigation was conducted in the three centers, including National Taiwan University Hospital (NTUH, 2200 beds), located in northern Taiwan, that serves an estimated civilian population of 2.6 million mainly from Taipei city, with annual carbapenems, piperacillin/tazobactam and tigecycline consumption rates of about 45, 40, and 3.0 defined daily dose (DDD) per 1000 patient-days respectively [14]; Chi-Mei Medical Center (CMMC, leading site, 1278 beds, with annual carbapenems, piperacillin/tazobactam, and tigecycline consumption rates of about 22, 40, and 8.0 DDD/1000 patient-days respectively) and National Cheng Kung University Hospital (NCKU, 1193 beds, with annual carbapenems, piperacillin/tazobactam, and tigecycline consumption rates of about 35, 35, and 2.0 DDD/1000 patient-days respectively). Both CMMC and NCKU are located in southern Taiwan, serving an estimated civilian population of 1.8 million mainly from Tainan city.

### 2.4. Assessment of Outcomes

#### 2.4.1. Primary Clinical Outcomes at the End of Therapy (EOT)

Cure was defined as the resolution of clinical signs and symptoms of infections compared with baseline, without a requirement for additional antibacterial treatment. Improvement was defined as the partial resolution of clinical signs and symptoms, with a scope of additional antibacterial treatment. Failure was defined as adding or switching to new antibiotics due to poor response, persistence, worsening signs and symptoms, unrelenting signs and symptoms, or persistent isolation of pathogens from the infection sites. Indeterminate was defined as the outcome that could not be assessed for any reason or adding/switching to new antibiotics due to emerging infections (other than initial infection sites) [5,6,15,27].

#### 2.4.2. Secondary Clinical Outcomes by 30 Days after EOT

Mortality referred to overall death and was assessed on the 30th day after the end of therapy (EOT). Clinical success was stringently defined survival without readmission between EOT and the 30th day after EOT. Readmission could be a potential outcome confounder [28,29].

#### 2.4.3. Microbiological Outcomes

Follow-up cultures of sampling from primary sites of infections were taken based on routine clinical practice. Eradication was defined as a documented disappearance of baseline infecting pathogens at primary sites of infections from the follow-up cultures. Persistence was defined as documented positive follow-up cultures from primary infection sites. If follow-up cultures were not available, it could be arbitrarily presumed to be eradication based on clinical success (cure or improvement) or to be persistence through clinical failure [6]. Superinfection was defined as the emergence of a new isolate at the primary site of infection with worsening signs and symptoms of infection.

#### 2.4.4. Data Collection

The collected data included demography, comorbidity, dates of admission and onset, source of infection, infection site, symptoms, severity status including the worst physiological variables of the baseline Acute Physiology and Chronic Health Evaluation (APACHE) II score within 48 h of intensive care unit (ICU) admission and real-time Sequential Organ Failure Assessment (SOFA) score during the first 48 h after infection onset, infectious complication, pathogen, empirical antimicrobial therapy, the date of end of therapy, outcome assessment, and the days of hospital stay after the onset of infection. Infections were classified as nosocomial vs. community-acquired depending on the onset of the infection after or before 48 h of hospitalization. Inadequate empirical antibiotic therapy was defined to be greater than 48 h between the time of culture and initiation of treatment using an antimicrobial for which the target pathogen was ultimately determined to be resistant in vitro [15].

### 2.5. Statistical Analyses

The baseline characteristics of selected patients from each hospital were analyzed using Pearson’s Chi-square test or Fisher’s exact test for categorical variables and analysis of variance (ANOVA) or Student’s *t*-test for continuous variables. The Kaplan–Meier plot was employed to present the trend on 30-days mortality after infection onset, and the trend differences between the three hospitals were compared using the long-rank test.

The Cox proportional hazard regression analysis through three steps was used to estimate the relative 30-day mortality risk after EOT. Firstly, the above-analyzed significant baseline characteristics, including symptoms, comorbidity, disease severity, and complications, and other potentially confounding variables were further screened by the univariate Cox proportional hazards regression analysis. Secondly, all variables with *p*-values < 0.05 from the univariate analysis were included in the first multivariable prediction model. Thirdly, the second multivariable prediction model used Collett’s model selection approach to find further predictors. This approach used a significance level of 0.10 for the univariate screening, after which the predictors with entry and stay criteria of 0.05 were selected in the final model. All statistical analyses were performed using SAS (version 9.4 for Windows, SAS Institute Inc., Cary, NC, USA). Significance was set at *p*-value < 0.05 (two-tailed).

### 2.6. Ethics Approval 

The study was approved by local ethics committees, and was exempted from the requirement for informed consent by the Institutional Review Board (IRB) of the National Taiwan University Hospital (IRB no. 201710028RINB), Taipei; the IRB of Chi Mei Medical Center (IRB no. 10511-002) and the IRB of National Cheng Kung University Hospital (IRB no. A-ER-106-221), Tainan, Taiwan.

## 3. Results

A total of 71 cases that received tigecycline-based therapy for ESBL-producing Enterobacteriaceae with or without carbapenem resistance were enrolled in the study. Among them, 16 were from NTUH, 17 from CMMC, and 38 from NCKU (Table 1). The infections were predominantly acquired at hospitals (90.1%). CMMC had more community-acquired cases (35.3%, *p* < 0.0001) and the lowest SOFA score (mean, 6.2, *p* = 0.033). The patients from NTUH had higher rates of post-operation status, source control, carbapenem resistance, septic shock, prolonged prothrombin time, and acute respiratory distress syndrome (ARDS) with statistical significance.

### 3.1. Patient Characteristics

The proportion of female patients in CMMC was significantly lower than others (*p* = 0.04). The patients from NTUH were significantly older in age (mean age, 72.1 years) than those from the other two hospitals (*p* = 0.02). The comorbidities of patients from NTUH involved significantly more respiratory disease, and solid cancer, with a higher Charlson score (mean, 9.5) than other hospitals. Other underlying diseases did not show significant differences among three institutes. No hematological malignancy or other immunocompromised hosts were found in the study. Most patients had multiple comorbidities with a mean Charlson score >5, most commonly cardiovascular disease (64.8%), metabolic diseases (57.8%), and solid cancer (42.3%). The patients from NTUH had a significantly higher rate of respiratory symptoms (dyspnea and cough) at the onset of infection. Most subjects were critically ill with a mean APACHE II score of more than 20, with 77.5% in the ICUs and 62.0% using mechanical ventilator (Table 1).

### 3.2. Infection Sites and Characteristics

The distribution of infection sites was similar, including cIAI (39.4%), cSSTI (36.6%), pneumonia (32.4%), and rarer in UTI (19.7%). Multiple infection foci were found in 25% of patients. NCKU had significantly fewer cases of pneumonia and UTI, whereas NTUH had a higher proportion of pneumonia and UTI (Table 1). Carbapenem resistance occurred in about 50% in each infection site (cSSTI, 50%; cIAI, 57%; and pneumonia, 48%). KP (ESBL-CR) was the most common pathogen in patients with cSSTI (46.2%), cIAI (50%), and pneumonia (43.5%; Table 2).

### 3.3. Pathogens

The ESBL-producing pathogens included 39 *K. pneumoniae* isolates (28 with carbapenem resistance), 30 *E. coli* isolates (3 with carbapenem resistance), 4 *Enterobacter cloacae* isolates (2 with carbapenem resistance), and 1 *Citrobacter freundii*. Four patients had concurrent *E. coli* and *K. pneumoniae* infection.

Nearly all patients received combination therapy, mostly with carbapenems (meropenem, imipenem, or doripenem), followed by cefepime, ceftazidime, piperacillin-tazobactam, cefpirome, piperacillin, and rarely, ciprofloxacin or levofloxacin. The rate of inappropriate empirical antimicrobial therapy was highly distributed among three hospitals without any significant difference (mean, 84.5%, Table 1). The remarkably high rate of inadequate empirical antibiotic therapy in three centers was due to a high proportion of carbapenem resistance and the unexpected presence of ESBL.

The outcome revealed that the patients from NTUH had significantly lower rates of clinical success (43.8%) and microbiological eradication (0%) at EOT, accompanied by the lowest survival rate by 30 days after EOT (37.5%). However, survival duration analysis for the 30-day mortality of the population among the three centers using the Kaplan–Meier technique showed no noteworthy difference between each of the paired comparisons (Figure 1).

Overall, the rates of clinical success and microbiological eradication at EOT were 63.4% and 50.7%, respectively. The overall mortality by 30 days after EOT was 36.6%. The mean length of stay at the hospital after the infection was 44.7 days, which was not significantly variable in any of the three hospitals (Table 1).

During the study period, the tigecycline susceptibility test was optional upon a physician request. We only can enroll tigecycline-susceptible cases (probably the physician would use tigecycline based on susceptibility), but persistent isolates at the end of therapy usually lacked tigecycline susceptibility (probably the physician switched to other drugs and did not request tigecycline testing). We could not say the development of tigecycline resistance explaining the cases with failure of microbiological eradication. From very limited cases with persistent isolates doing tigecycline susceptibility testing, tigecycline remained susceptible at the end of therapy.

The clinical outcomes of tigecycline-based treatment per patient subgroup by infection sites are shown in Table 2. The cSSTI, cIAI, and pneumonia were successfully treated in 61.5%, 60.7%, and 52.2% patients at EOT respectively. We did not evaluate the detailed outcomes of the UTI, because tigecycline was not primarily used to treat UTI, and 11 of 14 UTIs (78.6%) were accompanied infections with either cSSTI, cIAI, and/or pneumonia. The UTI and multiple infections were successfully treated in 57.1% and 44.4% patients at EOT respectively.

For cSSTI, the clinical and microbiological outcomes at EOT and survival 30 days post EOT, the ESBL-CS group had apparently higher clinical success rates (about 60–70%) than ESBL-CR group (about 20–50%), which were statistically insignificant. Meanwhile, the *E*. *coli* group had a better clinical success than the *K. pneumoniae* group at EOT (90% vs. 50%, *p* = 0.08) and by 30 days after EOT (90% vs. 36%, *p* = 0.01).

For cIAI, the clinical and microbiological outcomes at EOT and 30 days after EOT, either by *E. coli* or *K. pneumoniae* (ESBL-CS or ESBL-CR), were about 40–60% successful, holding no significance.

For pneumonia, the ESBL-CS group had higher clinical success rates than ESBL-CR at EOT (69.2% vs. 36.4%) and by 30 days after EOT (50.8% vs. 27.3%), although not statistically significant. The *E. coli* group showed a significantly higher clinical success rate than *K. pneumoniae* group at EOT (80% vs. 36%, *p* = 0.047). The microbiological outcomes of pneumonia revealed significant success in eradicating all ESBL-CS strains compared to all ESBL-CR ones (69.2% vs. 0%, *p* = 0.0006), especially in *K. pneumoniae* (ESBL-CS) versus *K. pneumoniae* (ESBL-CR) groups (75% vs. 0%, *p* = 0.011).

### 3.4. Risk Factors for Poor Clinical Outcome

The characteristic of the subjects with clinical success (survival without readmission) or mortality by 30 days after EOT was evaluated using univariate analysis (Table 3). The statistically significant predictors for mortality were found to be dyspnea, severity status (stay in ICU, ventilator use and SOFA score), and infectious complications (septic shock, acute renal failure, thrombocytopenia, prolonged prothrombin time, and neurologic complications). The clinical and microbiological outcomes at EOT were also correlated with clinical success by 30 days after EOT using both models (*p* < 0.0001). The mean length of stay after the infection was longer in the survival group than that in the mortality group (52.1 vs. 32.9, *p* = 0.03). The above-mentioned significant patient characteristics, microbiological eradication and other potential confounders, such as hospitals, Charlson score, carbapenem resistance, antibiotic therapy, and source control were further forwarded to the Cox proportional hazards regression analysis.

### 3.5. Hazard Ratio for Outcome

Table 4 shows independent factors associating with mortality by using Cox proportional hazards regression model. Univariate analysis revealed that dyspnea (*p* = 0.003), SOFA score (*p* = 0.0004), SOFA score >8 (*p* = 0.001), septic shock (*p* = 0.022), thrombocytopenia (*p* = 0.0004), prolonged prothrombin time (*p* = 0.018), and lack of microbiological eradication ability (*p* = 0.004) were considerably associated with mortality by 30 days after EOT.

Cox regression proportional hazard model revealed that only dyspnea (*p* = 0.008–0.009) and SOFA score > 8 (*p* = 0.041) were associated with the time to death. The prolonged prothrombin time and lack of microbiological eradication were found to be inconsistently associated with mortality when different approaching models were used. The Collett’s model selection approach revealed that only dyspnea (*p* = 0.0002) and SOFA > 8 score (*p* = 0.004) significantly predicted mortality, with nearly 7.3-fold and 4.6-fold increased odds of dying respectively. Therefore, no specific factors of carbapenem resistance, empirical antimicrobial therapy, source control, ventilator use, ICU stay, or hospitals were associated with time to mortality (Table 4), even though NTUH had a relatively higher number of cases with source control (Table 1).

## 4. Discussion

Our multicenter study focused on the tigecycline effectiveness versus infections caused by ESBL-producing *Enterobacteriaceae* susceptible or resistant towards carbapenems, in an almost evenly distributed infection sites of cIAI (39.4%), cSSTI (36.6%), and pneumonia (32.4%). The clinical success and microbiological eradication rates at EOT of tigecycline-based therapy were 63.4% and 50.7% respectively. The mean clinical success rate was 57.8% by 30 days after EOT, including cSSTI (57.7%), cIAI (57.1%), and pneumonia (43.5%).

In our study, *K. pneumoniae* (ESBL-CR) was the most common isolate (39.4% of the patients), followed by *E. coli* (ESBL-CS, 38.0%), *K. pneumoniae* (ESBL-CS, 15.5%), and *E. coli* (ESBL-CR, 4.2%). The *E. coli-*infected group (regardless of carbapenem resistance) had an overall higher clinical success or survival rate than *K. pneumoniae-*infected group. A large study of patients with bacteremia in 12 countries reported ESBL-producing *K. pneumoniae* contributed significantly more mortality rate than the ESBL-producing *E. coli* (34% vs. 17%) [30]. In similar, *E. coli* has a better survival rate than *Klebsiella* species (75% vs. 23%) in adults with community-onset bacteremia [23].

The effect of tigecycline-based therapy was lower than a previous report demonstrating a high overall clinical success rate (91.0%, 61/67) for ESBL-related infections [5]. The difference in the clinical success rate between our study and the previous report was statistically significant (63.4% vs. 91.0%, *p* = 0.0001). In another large series of cIAI patients, treatment was successful in 426 (70.3%) of 606 tigecycline-treated patients [28]. The difficulty of our study in attaining a high success rate could perhaps be justified by the data of *K. pneumoniae* (ESBL-CR) and a high proportion (80.5%) with inappropriate empirical antimicrobial therapy.

Carbapenem resistance is more common among infections caused by the Enterobacteriaceae and was associated with a significantly higher risk of mortality in comparison to carbapenem-susceptible Enterobacteriaceae [31].

In our previous study, we demonstrated that tigecycline activity was unaffected by ESBL production and had an MIC_90_ ranging from 0.5 to 1.5 μg/mL for ESBL-producing *K. pneumoniae*, *Serratia marcescens,* and *E. cloacae* [32]. In a total of 2870 blood samples in India, all ESBL producers were sensitive to tigecycline [33]. In a small collection of serious infections by *K. pneumoniae*, the cure rate for tigecycline in patients with multidrug-resistant *K. pneumoniae* was 5/6 (83.3%). All six patients achieved microbiological eradication [6]. Nonetheless, due to the high frequency of carbapenem resistance in infections of ESBL-producing *Enterobacteriaceae*, the treatment of choice for these ESBL-related infections should not always consider a carbapenem as an empirical regimen. The tigecycline-based therapy should be efficiently developed ahead to treat these infections empirically rather than just reserving it as a specific therapy at a later stage.

Bloodstream infections caused by ESBL-producing *Enterobacteriaceae* have been associated with higher mortality than those by non-ESBL producers [23,34]. This was also seen in our patients having pneumonia or cSSTI. The success rates seemed to be lower in patients with pneumonia caused by *K. pneumoniae* (50% of ESBL-CS and 30% of ESBL-CR groups), due to difficulty in the microbiological eradication of *K. pneumoniae* (ESBL-CR) in cases of pneumonia (0/10). Our results add evidence that tigecycline is not recommended to treat pneumonia by *K. pneumoniae* (ESBL-CR) strains.

A higher concentration of tigecycline or combination of tigecycline with colistin rendering good in vitro synergistic activity against ESBL-producing *K. pneumoniae* or with carbapenem resistance could be further reassessed for clinical success [35,36,37]. The combination of tigecycline with colistin, aminoglycoside or doxycycline might be active against the ESBL-CR infections in several studies in Taiwan [37,38,39].

Most of our patients (78%) were critically ill and stayed in the ICUs. Although the univariate analysis revealed several risk factors for mortality, including dyspnea, SOFA score > 8, septic shock, thrombocytopenia, prolonged prothrombin time, and inability towards microbiological eradication, the Cox regression proportional model demonstrated that only dyspnea and SOFA score or SOFA > 8 were independently associated with the estimated time towards death. The SOFA score has been a useful predictor of clinical outcome and an initial score of 8–11 was associated with a 60% ICU mortality [40]. About 82% of the patients with dyspnea used the respiratory ventilator. A further study may be required to investigate the relationship of outcome and various ventilator modes. Our study proposes that late-onset mortality could be predicted by the useful warning sign of dyspnea and a SOFA score, which represents real-time disease severity during the infection onset.

## 5. Conclusions

The limitations of this study included retrospective observational design, a lack of information about tigecycline side effects and drug interactions, a lack of information about development of tigecycline resistance, and the low number and its heterogeneity in demographics. However, the infection sites of cohort were almost evenly distributed. The clinical experience of tigecycline in the ESBL-related infections was limited in current practice. This multicenter study enrolling a sizeable sample of patients provides an evident basis of future generalization in the design of tigecycline-based strategies against the critically ill infections by ESBL producers. The tigecycline-based therapy for ESBL-producers was highly active (80–90% success rates) against pneumonia and cSSTI by *E. coli*, but showed moderate response (50–60% success rates) for cIAI caused by *E. coli* or *K. pneumoniae*. Microbiological eradication of *K. pneumoniae* with carbapenem resistance was difficult to achieve in pneumonia, for which tigecycline-based therapy should not be recommended. The relative effects of tigecycline might be cautiously integrated into ASP recommendation as an alternative regimen in future. Some variables could be useful to predict outcomes of critically ill patients. High SOFA score and dyspnea were independently associated with time till death even when the patients were put on ventilator.

## Figures and Tables

**Figure 1 antibiotics-09-00231-f001:**
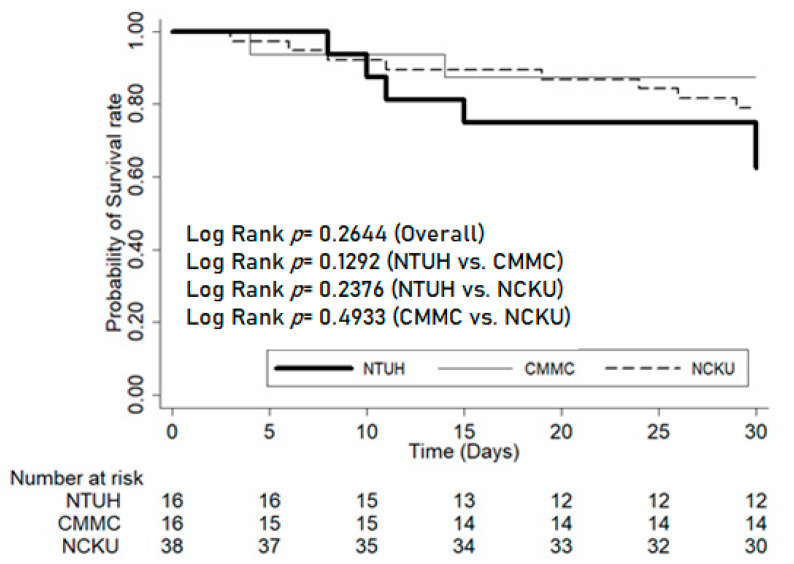
Kaplan–Meier plot; note: survival trend of the 30-day mortality among 3 hospitals using the Kaplan–Meier approach shows no significant difference between each population upon comparison. The digits below the chart were surviving patient numbers of each institute at each point of time. One patient’s outcome was indeterminate in CMMC and was not enrolled in the survival analysis.

**Table 1 antibiotics-09-00231-t001:** Demographic data, comorbidity, symptoms, infectious status, pathogens, and outcome of the patients who received tigecycline-based therapy for ESBL-producing *Enterobacteriaceae* with or without carbapenem resistance in 3 medical centers in Taiwan.

Variables	NTUH (*n* = 16)	CMMC (*n* = 17)	NCKU (*n* = 38)	*p*	Total (*n* = 71)
Sex, no (%)					
Female	5 (31.3)	3 (17.7)	20 (52.6)	0.04	28 (39.4)
Male	11 (68.8)	14 (82.4)	18 (47.4)		
Age (mean ± SD)	72.1 ± 15.4	66.3 ± 19.7	59.9 ± 13.8	0.02	64.2 ± 16.3
Underlying diseases, no (%)					
Cardiovascular disease	10 (62.5)	12 (70.6)	24 (63.2)	0.894	46 (64.8)
Respiratory disease	10 (62.5)	5 (29.4)	1 (2.7)	<0001	16 (22.5)
Neurological disease	6 (37.5)	5 (29.4)	5 (13.2)	0.09	16 (22.5)
Hepatobiliary disease	6 (37.5)	7 (41.2)	10 (26.3)	0.50	23 (32.4)
Metabolic disease	11 (68.8)	7 (41.2)	23 (60.5)	0.27	41 (57.8)
Autoimmune disease	2 (12.5)	0 (0.0)	1 (2.6)	0.18	3 (4.2)
Solid cancer	9 (56.3)	2 (11.8)	19 (50.0)	0.01	30 (42.3)
Charlson score (mean ± SD)	9.5 ± 3.5	2.8 ± 2.0	4.3 ± 2.2	<0001	5.1 ± 3.5
Community-acquired, no (%)	1 (6.3)	6 (35.3)	0 (0.0)	<0001	7 (9.9)
Symptoms on onset, no (%)					
Fever	7 (43.8)	11 (64.7)	14 (36.8)	0.16	32 (45.1)
Chest pain	0 (0.0)	1 (5.9)	1 (2.6)	0.72	2 (2.8)
Diarrhea	4 (25.0)	2 (11.8)	2 (5.3)	0.07	8 (11.3)
Dyspnea	13 (81.3)	7 (41.2)	2 (5.3)	<0001	22 (31.0)
Cough	6 (37.5)	2 (11.8)	0 (0.0)	0.0003	8 (11.3)
Infection site, no (%)					
cSSTI	9 (56.3)	3 (17.7)	14 (36.8)	0.08	26 (36.6)
cIAI	6 (37.5)	4 (23.5)	18 (47.4)	0.23	28 (39.4)
Pneumonia	10 (62.5)	10 (58.8)	3 (7. 9)	<0001	23 (32.4)
UTI	8 (50.0)	3 (17.6)	3 (7.9)	0.002	14 (19.7)
Severity status, no (%)					
Secondary bacteremia	5 (31.3)	2 (11.8)	3 (7.9)	0.07	10 (14.1)
Stay in ICU	12 (75.0)	14 (82.4)	29 (76.3)	0.87	55 (77.5)
Post operation status	9 (56.3)	8 (47.1)	2 (5.3)	<0001	19 (26.8)
Ventilator use	12 (75.0)	13 (76.5)	19 (50.0)	0.10	44 (62.0)
SOFA score (mean ± SD)	7.8 ± 2.8	6.2 ± 5.1	9.3 ± 4.2	0.03	8.2 ± 4.3
APACHE II score (mean ± SD)	20.3 ± 9.9	23.5 ± 9.9	- ^a^	0.46	- ^a^
Complications, no (%)					
Septic shock	13 (81.3)	7 (41.2)	18 (47.4)	0.04	38 (53.5)
Acute renal failure	10 (62.5)	7 (41.2)	18 (47.4)	0.46	35 (49.3)
Liver function impairment	4 (25.0)	6 (35.3)	25 (65.8)	0.01	35 (49.3)
Platelet count <100,000/μL	6 (37.5)	5 (29.4)	15 (39.5)	0.80	26 (36.6)
Prolonged prothrombin time	8 (50.0)	3 (17.7)	2 (5.3)	0.0006	13 (18.3)
ARDS	10 (62.5)	0 (0.0)	0 (0.0)	<0001	10 (14.1)
Neurological complication	3 (18.8)	0 (0.0)	13 (34.2)	0.009	16 (22.5)
Pathogen, no (%)					
*Escherichia coli*	2	14	14		30
With carbapenem resistance	2 (100.0)	0 (0.0)	1 (7.1)	0.007	3 (10.0)
*Klebsiella pneumoniae*	14	3	22		39
With carbapenem resistance	14 (100.0)	0 (0.0)	14 (63.6)	<0001	28 (71.8)
Empirical Antibiotic therapy					
Inappropriate	16 (100.0)	13 (76.5)	31 (81.6)	0.1	60 (84.5)
Therapy for infection, no (%)					
Antibiotic only	5 (31.3)	12 (70.6)	31 (81.6)	0.001	48 (67.6)
Antibiotic + source control	11 (68.8)	5 (29.4)	7 (18.4)		23 (32.4)
Clinical outcome on EOT					
Success (cure/improvement)	7 (43.8)	14 (82.4)	24 (63.2)	0.02	45 (63.4)
Failure	6 (37.5)	3 (17.7)	14 (36.8)		23 (32.4)
Undetermined	3 (18.8)	0 (0.0)	0 (0.0)		3 (4.2)
Microbiological outcome on EOT					
Eradication	0 (0.0)	12 (70.6)	24 (63.2)	<0001	36 (50.7)
Clinical outcome on 30 days after EOT ^b^					
Success (survival, no readmission)	6 (37.5)	12 (75.0)	23 (60.5)	0.04	41 (57.8)
Overall death	8 (50.0)	3 (18.8)	15 (39.5)		26 (36.6)
Readmission	2 (12.5)	1 (6.3)	0 (0.0)		3 (4.2)
Days on discharge after infection (mean ± SD)	44.7 ± 39.7	35.7 ± 25.6	48.7 ± 46.9	0.86	44.7 ± 40.9

Note: cSSTI, complicated skin and skin tissue infections; cIAI, complicated intra-abdominal infections; UTI, urinary tract infection; ICU, intensive care unit; SOFA, Sequential Organ Failure Assessment; APACHE, Acute Physiology and Chronic Health Evaluation; ARDS, acute respiratory distress syndrome; ^a^ Data were incomplete in National Cheng Kung University Hospital (NCKU); EOT, end of therapy; ^b^ One patient data was indeterminate in Chi-Mei Medical Center (CMMC).

**Table 2 antibiotics-09-00231-t002:** EOT outcome and success (survival without readmission) by 30 days after EOT for patients with cSSTI, cIAI or pneumonia stratified by pathogens.

Bacteriology	Success at EOT (Cure/Improvement)	*p*	Microbiological Eradication at EOT	*p*	Success by 30 Days after EOT	*p*
cSSTI (*n* = 26) ^a^	16 (61.5%)		11 (42.3%)		15 (57.7%)	
ESBL-CS (*n* = 13)	9 (69.2%)	0.69	8 (61.5%)	0.11	10 (76.9%)	0.11
ESBL-CR (*n* = 13)	7 (53.8%)		3 (23.1%)		5 (38.5%)	
*Escherichia coli* (*n* = 10)	9 (90%)		7 (70%)		9 (90%) ^b^	
ESBL-CS (*n* = 9)	8 (88.9%)		7 (77.8%)		8 (88.9%)	
ESBL-CR (*n* = 1)	1 (100%)		0 (0%)		1 (100%)	
*Klebsiella pneumoniae* (*n* = 14)	7 (50%)		4 (28.6%)		5 (35.7%) ^b^	
ESBL-CS (*n* = 2)	1 (50%)		1 (50%)		1 (50%)	
ESBL-CR (*n* = 12)	6 (50%)		3 (25%)		4 (33.3%)	
cIAI (*n* = 28) ^c^	17 (60.7%)		13 (46.4%)		16 (57.1%)	
ESBL-CS (*n* = 13)	7 (53.8%)	0.22	6 (46.2%)	0.90	6 (46.2%)	0.38
ESBL-CR (*n* = 16)	10 (62.5%)		7 43.8%)		10 (62.5%)	
*Escherichia coli* (*n* = 10)	7 (60%)		5 (50%)		6 (60%)	
ESBL-CS (*n* = 8)	5 (62.5%)		4 (50%)		4 (50%)	
ESBL-CR (*n* = 2)	2 (100%)		1 (50%)		2 (100%)	
*Klebsiella pneumoniae* (*n* = 19)	9 (47.4%)		7 (36.8%)		9 (47.4%)	
ESBL-CS (*n* = 5)	2 (40%)		2 (40%)		2 (40%)	
ESBL-CR (*n* = 14)	7 (50%)		5 (35.7%)		7 (50%)	
Pneumonia (*n* = 23) ^d^	12 (52.2%)		9 (39.1%)		10 (43.5%)	
ESBL-CS (*n* = 13)	9 (69.2%)	0.22	9 (69.2%)	0.0006	7 (53.8%)	0.24
ESBL-CR (*n* = 11)	4 (36.4%)		0 (0%)		3 (27.3%)	
*Escherichia coli* (*n* = 10)	8 (80%) ^e^		6 (60%)		6 (60%)	
ESBL-CS (*n* = 9)	7 (77.8%)		6 (66.7%)	0.40	5 (55.6%)	
ESBL-CR (*n* = 1)	1 (100%)		0 (0%)		1 (100%)	
*Klebsiella pneumoniae* (*n* = 14)	5 (35.7%) ^e^		3 (21.4%)		4 (28.6%)	
ESBL-CS (*n* = 4)	2 (50%)		3 (75%)	0.011	2 (50%)	
ESBL-CR (*n* = 10)	3 (30%)		0 (0%)		2 (20%)	
Multiple infections (*n* = 18)	8 (44.4%)		4 (22.2%)		6 (33.3%)	

Note. EOT, end of therapy; cSSTI, complicated skin and skin tissue infections; ESBL-CS, ESBL with carbapenem susceptibility; ESBL-CR, ESBL with carbapenem resistance; ^a^ Two cases were infected *Enterobacter cloacae*; ^b^ 90% vs. 35.7%, *p* = 0.01; cIAI, complicated intra-abdominal infection; ^c^ Three cases had both *E. coli* and *K. pneumoniae* and two cases were infected *E. cloacae*; ^d^ One case was infected by both *Escherichia coli* and *Klebsiella pneumoniae*; ^e^ 80% vs. 35.7%, *p* = 0.047.

**Table 3 antibiotics-09-00231-t003:** Predictors for the clinical outcome by 30 days after the end of therapy (EOT) ^a.^

Variables, no. (%)	Success (Survival without Readmission) (*n* = 41)	All-Cause Death (*n* = 26)	*p*
Sex, no (%)			
Female	13 (31.7)	14 (53.8)	0.08
Male	28 (68.3)	12 (46.2)	
Age	63.1 ± 16.4	64.7 ± 16.6	0.93
Acquired source of infection			
Community	5 (12.2)	1 (3.9)	0.15
Hospital and healthcare institute	36 (87.8)	25(96.2)	
Symptoms on onset			
Fever	21 (51.2)	8 (30.8)	0.13
Dyspnea	7 (17.1)	12 (46.2)	0.01
Cough	3 (7.3)	5 (19.2)	0.25
Chest pain	2 (4.9)	0 (0.0)	0.52
Diarrhea	5 (12.2)	2 (7.7)	0.70
Infection site			
cSSTI	15 (36.6)	9 (34.6)	>999
cIAI	16 (39.0)	12 (46.2)	0.62
Pneumonia	10 (24.4)	10 (38.5)	0.28
UTI	5 (12.2)	6 (23.1)	0.32
Severity status			
Secondary bacteremia	6 (14.6)	4 (15.4)	>999
Stay in ICU	28 (68.3)	24 (92.3)	0.03
Post operation status	12 (29.3)	4 (15.4)	0.25
Ventilator use	21 (51.2)	20 (76.9)	0.04
SOFA score	6.71 ± 3.7	10.77 ± 4.3	0.0004
APACHE II score	21.4 ± 10.0	20.0 ± 8.8	0.77 ^b^
Underlying diseases			
Cardiovascular disease	28 (68.3)	16 (61.5)	0.61
Respiratory disease	9 (22.5)	6 (23.1)	>999
Neurological disease	8 (19.5)	5 (19.2)	>999
Hepatobiliary disease	16 (39.0)	6 (23.1)	0.20
Metabolic disease	27 (65.9)	12 (46.2)	0.13
Autoimmune disease	1 (2.4)	2 (7.7)	0.56
Solid cancer	17 (41.5)	11 (42.3)	>999
Charlson score (mean ± SD)	4.6 ± 2.8	5.6 ± 4.1	0.68
Complications			
Septic shock	18 (43.9)	19 (73.1)	0.03
Acute renal failure	16 (39.0)	17 (65.4)	0.047
Liver function impairment	19 (46.3)	16 (61.5)	0.32
Platelet count < 100,000/μL	8 (19.5)	17 (65.4)	0.0002
Prolonged prothrombin time	4 (9.8)	8 (30.8)	0.048
ARDS	4 (9.8)	5 (19.2)	0.29
Neurological complication	6 (14.6)	10 (38.5)	0.04
Pathogen, no (%)			
*Escherichia coli* (ESBL-CR)	3 (14.3)	0 (0.0)	0.55
*Klebsiella pneumoniae* (ESBL-CR)	12 (66.7)	14 (73.7)	0.73
Therapy for infection, no (%)			
Appropriate empirical antibiotic therapy	6 (14.6)	5 (19.2)	0.74
Antibiotic + source control	15 (36,6)	6 (23.1)	0.25
Clinical outcome on EOT			
Success (cure/improvement)	38 (92.7)	3 (11.5)	<0001
Failure	3 (7.32)	20 (76.92)	
Undetermined	0 (0.00)	3 (11.54)	
Microbiological outcome on EOT			
Eradication	29 (70.7)	5 (19.2)	<0001
Days on discharge after infection (mean ± SD)	52.1 ± 46.79	32.9 ± 29.3	0.03

Note. ARDS, acute respiratory distress syndrome; ESBL-CR, ESBL with carbapenem resistance; EOT, end of therapy; ^a^ excluded one patient of indeterminate in CMMC and 3 patients with readmission; ^b^ Data were incomplete in NCKU.

**Table 4 antibiotics-09-00231-t004:** Hazard ratio of 30 days mortality after end of therapy using the Cox regression model.

Variables	Heading	Heading	Model 1 (*p* < 0.05 with Categorical)		Model 2 (Collett’s Model Selection Approach)	
	Univariate HR	*p*	Multivariable HR	*p*	Multivariable HR	*p*
	(95% C.I.)		(95% C.I.)		(95% C.I.)	
Hospital						
NTUH	reference					
CMMC	0.31 (0.06–1.53)	0.150				
NCKU	0.53 (0.19–1.53)	0.243				
Dyspnea symptom	4.71 (1.71–13.00)	0.003	14.49 (2.01–104.48)	0.008	7.33 (2.58–20.78)	0.0002
Stay in ICU	2.27 (0.52–10.00)	0.278				
Ventilator use	2.17 (0.70–6.73)	0.180				
SOFA score	1.22 (1.09–1.36)	0.0004				
SOFA score (categorical)						
≤8	reference		reference		reference	
>8	7.95 (2.26–27.96)	0.001	10.04 (1.10–92.00)	0.041	4.58 (1.64–12.79)	0.004
APACHE II score	1.01 (0.94–1.08)	0.812				
APACHE II score (categorical)						
≤20	reference					
>20	1.32 (0.46–3.79)	0.612				
Charlson score	1.09 (0.96–1.24)	0.187				
Charlson (categorical)						
0	reference					
1	0.88 (0.06–14.10)	0.929				
≥2	0.78 (0.10–5.97)	0.815				
Complications						
Septic shock	4.35 (1.24–15.29)	0.022	1.68 (0.23–12.38)	0.612		
Acute renal failure	2.52 (0.87–7.25)	0.088				
Thrombocytopenia	9.70 (2.75–34.15)	0.0004	0.47 (0.04–5.82)	0.558		
Prolonged prothrombin time	3.39 (1.23–9.35)	0.018	0.28 (0.04–2.06)	0.210		
ARDS	2.21 (0.71–6.85)	0.170				
Neurological complication	1.69 (0.59–4.87)	0.331				
Microbiological eradication	0.11 (0.03–0.50)	0.004	0.16 (0.03–0.93)	0.042		
*Klebsiella pneumoniae*						
ESBL-CS	reference					
ESBL-CR	0.71 (0.21–2.35)	0.574				
Therapy for infections						
Antibiotic only	reference					
Antibiotic + source control	0.43 (0.10–1.91)	0.269				
Empirical antibiotic therapy						
Appropriate	reference					
Inappropriate	0.52 (0.17–1.61)	0.255				

Note. ICU, intensive care unit; ARDS, acute respiratory distress syndrome; SOFA, Sequential Organ Failure Assessment; APACHE, Acute Physiology and Chronic Health Evaluation; ESBL-CS: ESBL with carbapenem susceptibility; ESBL-CR: ESBL with carbapenem resistance.

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
