# Peer review of "Tigecycline Therapy for Infections Caused by Extended-Spectrum β-Lactamase-Producing Enterobacteriaceae in Critically Ill Patients"

_antibiotics, 2020, doi:10.3390/antibiotics9050231_

Round 1
Reviewer 1 Report
Content comments:
- Did the authors notice any side effects of tigecycline ? Did the authors need to reduce the dose due to them ?
- Could the authors exclude any drug interactions ? They do not mention the inclusion criteria of the patients regarding the concomitant medication of the patients.
Based on the above-mentioned facts, from my point of view, the article could be published after incorporation of the responses to the questions above.
Author Response
To Reviewer 1 comments:
- Did the authors notice any side effects of tigecycline? Did the authors need to reduce the dose due to them?
R: Thanks for the important questions. However, this retrospective study was primarily to evaluate the effect of tigecycline and secondly to analyze the clinical characteristics predictive of mortality. The initially designed case collected form did not notice tigecycline side effects and dose adjustment. At current stage, we could not offer these drug-related issues.
- Could the authors exclude any drug interactions? They do not mention the inclusion criteria of the patients regarding the concomitant medication of the patients.
R: Thanks for the important questions. The initially designed case form did not collect information of drug interactions. Patients did receive concomitant medication. The enrolled patients completed their course to evaluate clinical and microbiological outcomes. At current stage, we could not offer the drug-interaction information.
- P351-353, We only could list limitation of the study including lack of information about tigecycline side effects and drug interactions.

Reviewer 2 Report
Yu et al. objective was to evaluate of tigecycline in the treatment of serious infections caused by ESBL-producing Enterobacteriaceae in a health-care setting, with the aim to advise future therapeutic strategies. Authors described the impact of tigecycline use in a cohort of patients with cSSTI, cIAI, UTI and pneumonia, nested within a multicentre, retrospective study. This is an important public health issue and data from different countries, where the AMR burden is higher, are surely needed.
The study shows two main drawbacks. One is that the Authors did not confirm the susceptibility of isolated bacterial strains to tigecycline in vitro, the other is Table 2, in which data for UTI was not included. Furthermore, it would have been nice to present separately the cases associated with Enterobacter infections as well as the mixed infections, in this manuscript. Regarding to the first suggestion: would be possible to the Authors to add the results of test tigecycline susceptibility? What was the tigecycline susceptibility of strains isolated at the end of therapy? What can you say about the development of tigecycline resistance in these patients? These data are obligatory because they can affect the discussion/conclusions made regarding the epidemiology of the resistant microorganisms as well as any potential of these data to be used as reference in therapy.
Minor comments for Authors consideration:
Lines 49 and 94: Please double-check the whole manuscript for references to the literature.
Line 198: Please double-check the whole manuscript for abbreviations, e.g. instead of SSTI should be cSSTI.
Fig 1: What does the digits below the chart?
Author Response
Reviewer 2 comments:
Yu et al. objective was to evaluate of tigecycline in the treatment of serious infections caused by ESBL-producing Enterobacteriaceae in a health-care setting, with the aim to advise future therapeutic strategies. Authors described the impact of tigecycline use in a cohort of patients with cSSTI, cIAI, UTI and pneumonia, nested within a multicentre, retrospective study. This is an important public health issue and data from different countries, where the AMR burden is higher, are surely needed.
The study shows two main drawbacks.
1. One is that the Authors did not confirm the susceptibility of isolated bacterial strains to tigecycline in vitro. Regarding to the first suggestion:
(1) would be possible to the Authors to add the results of test tigecycline susceptibility?
R: P90: All isolates were susceptible to tigecycline.
(2) What was the tigecycline susceptibility of strains isolated at the end of therapy?
R: Thanks for the important questions. This was a retrospectively study. P82-83: During the study period, tigecycline test was not routinely performed for ESBL-producing isolates, but was optional upon a choice of physicians.
P230-233: We only can enroll tigecycline-susceptible cases (probably physician would use tigecycline based on susceptibility), but persistent isolates at the end of therapy usually lacked of tigecycline susceptibility (probably physician switched to other drugs and did not request tigecycline testing).
(3) What can you say about the development of tigecycline resistance in these patients?
R: P234-235: We could not say the development of tigecycline resistance explaining the cases with failure of microbiological eradication. From very limited cases with persistent isolates doing tigecycline susceptibility testing, tigecycline remained susceptible at the end of therapy.
(4) These data are obligatory because they can affect the discussion/conclusions made regarding the epidemiology of the resistant microorganisms as well as any potential of these data to be used as reference in therapy.
R: Thanks for the important comments. The current retrospective study could not have the capability to evaluate the epidemiology of tigecycline resistance. We could add limitation of lack of information about development of tigecycline resistance in p351-353.
2. Table 2, in which data for UTI was not included.
R: Thanks for the questions. UTI is not an indication of tigecycline (as rare urinary excretion). Although we did not exclude the UTI cases, but we did not intend to analyze UTI as a single group.
P 238-241: We did not evaluate the detailed outcomes of the UTI, because tigecycline was not primarily used to treat UTI, and 11 of 14 UTIs (78.6%) were accompanied infections with either cSSTI, cIAI, and/or pneumonia. The UTI and multiple infections were successfully treated in 57.1% and 44.4% patients at EOT respectively.
3. Furthermore, it would have been nice to present separately the cases associated with Enterobacter infections as well as the mixed infections, in this manuscript.
R: Only 4 cases had Enterobacter infections and, if fact, Table 2 has been too complex enough, if adding more data like Enterobacter, UTI and mixed infections inside, it would be too difficult to read. As limited total case number, it would not be helpful to show too complex classification. We just add outcomes of multiple infection (Table 2 last line). P240-241: The UTI and multiple infections were successfully treated in 57.1% and 44.4% patients at EOT respectively.
Minor comments for Authors consideration:
1. Lines 49 and 94: Please double-check the whole manuscript for references to the literature.
R: We have work through the text to double-check the references.
2. Line 198: Please double-check the whole manuscript for abbreviations, e.g. instead of SSTI should be cSSTI.
R: Line 103: cSSTI, Line 201 cSSTI
3. Fig 1: What does the digits below the chart?
R: Those were interior data used to calculate the risk of time to death at any point of time in the Kaplan-Meier plot.
P223-224: The digits below the chart were surviving patient numbers of each institute at each point of time. One patient’s outcome was indeterminate in CMMC and was not enrolled in the survival analysis.

Reviewer 3 Report
Manuscript - Tigecycline Therapy for Infections Caused by 2 Extended-spectrum β-Lactamase-producing 3 Enterobacteriaceae in critically ill patients has been evaluated for publication in Antibiotics and publication of the paper is proposed after a minor revision. In general; In the present study, tigecycline was used in 71 patients in 3 medical centres in Taiwan. Primary, secondary clinical results, microbiological results and statistical analyses were examined, results and discussion are adequately well structured and written.
Comments: The authors are suggested to improve:
Lines 23 – 38, (Abstract): Abstract is including Objectives, Methods, Results and Conclusion, to my opinion Abstract should be rewritten without four mentioned sections.
Line 124, wrong literature citation [5,6.15.27].
Line 167 Sub-title: Institutes should be removed.
Line 175, population data on males (age, number) in the Table 1 are missing. Line 183-192, Patient characteristics, in this part I am missing comments on male/female characteristics/variation according to cardiovascular/metabolic diseases, solid cancer, etc.
Line 212. Title: Outcome stratified by institutes, and Line 225 Outcome stratified by infection sites should be modified.
Line 218, Figure 1, poor picture quality.
Line 259, population data on males (age, number) in the Table 3 are missing again. Line 342-349, This part is actalluy the Conclusions of the work/article. I therefore propose to add some more data and also to include the paragraph - lines 336-341 in a separate section - Conclusions.
Author Response
Reviewer 3 comments:
Manuscript - Tigecycline Therapy for Infections Caused by Extended-spectrum β-Lactamase-producing Enterobacteriaceae in critically ill patients has been evaluated for publication in Antibiotics and publication of the paper is proposed after a minor revision. In general; In the present study, tigecycline was used in 71 patients in 3 medical centres in Taiwan. Primary, secondary clinical results, microbiological results and statistical analyses were examined, results and discussion are adequately well structured and written.
Comments: The authors are suggested to improve:
- Lines 23 – 38, (Abstract): Abstract is including Objectives, Methods, Results and Conclusion, to my opinion Abstract should be rewritten without four mentioned sections.
R: We delete the sub-heading.
- Line 124, wrong literature citation [5,6.15.27].
R: Line 126: It was corrected to [5, 6, 15, 27]
- Line 167 Sub-title: Institutes should be removed.
R: Line 169: OK to be removed.
- Line 175, population data on males (age, number) in the Table 1 are missing.
R: Table 1. It is now added, due to the same variable of sex, the same p value as female).
- Line 183-192, Patient characteristics, in this part I am missing comments on male/female characteristics/variation according to cardiovascular/metabolic diseases, solid cancer, etc.
R: Lines-185-195: we have made major revision.
The proportion female patients in CMMC was significantly lower than others (p = 0.04). The patients from NTUH were significantly older in age (mean age, 72.1 years) than those from the other two hospitals (p = 0.02). … No hematological malignancy or other immunocompromised host were found in the study….
The comorbidities of patients from NTUH involved significantly more respiratory disease, and solid cancer…. Other underlying diseases did not show significant difference among three institutes.
- Line 212. Title: Outcome stratified by institutes, and Line 225 Outcome stratified by infection sites should be modified.
R: Line 215, Line 225: OK to be removed.
- Line 218, Figure 1, poor picture quality.
R: This has been the best quality we can made.
- Line 259, population data on males (age, number) in the Table 3 are missing again.
R: Line 272 (Table 3): It is now added, due to the same variable of sex, the same p value as female).
- Line 342-349, This part is actalluy the Conclusions of the work/article. I therefore propose to add some more data and also to include the paragraph - lines 336-341 in a separate section - Conclusions.
R: Line 350-365: Conclusion section has been built. Some more data were added.
Lines 356-358: The tigecycline-based therapy for ESBL-producers was highly active (80%-90% success rates) against pneumonia and cSSTI by E. coli, but showed moderate response (~50%–60% success rates) for cIAI caused by E. coli or K. pneumoniae.

Round 2
Reviewer 2 Report
I accept the manuscript in the present form.